# Generalized Bayesian Posterior Expectation Distillation for Deep Neural Networks

## Abstract

In this paper, we present a general framework for distilling expectations with respect to the Bayesian posterior distribution of a deep neural network, significantly extending prior work on a method known as "Bayesian Dark Knowledge." Our generalized framework applies to the case of classification models and takes as input the architecture of a "teacher" network, a general posterior expectation of interest, and the architecture of a "student" network. The distillation method performs an online compression of the selected posterior expectation using iteratively generated Monte Carlo samples from the parameter posterior of the teacher model. We further consider the problem of optimizing the student model architecture with respect to an accuracy-speed-storage trade-off. We present experimental results investigating multiple data sets, distillation targets, teacher model architectures, and approaches to searching for student model architectures. We establish the key result that distilling into a student model with an architecture that matches the teacher, as is done in Bayesian Dark Knowledge, can lead to sub-optimal performance. Lastly, we show that student architecture search methods can identify student models with significantly improved performance.

## 1 Introduction

Deep learning models have shown promising results in the areas including computer vision, natural language processing, speech recognition, and more (Krizhevsky et al., 2012; Graves et al., 2013a;b; Huang et al., 2016; Devlin et al., 2018). However, existing point estimation-based training methods for these models may result in predictive uncertainties that are not well calibrated, including the occurrence of confident errors.

It is well-known that Bayesian inference can often provide more robust posterior predictive distributions in the classification setting compared to the use of point estimation-based training. However, the integrals required to perform Bayesian inference in neural network models are also well-known to be intractable. Monte Carlo methods provide one solution to representing neural network parameter posteriors as ensembles of networks, but this can require large amounts of both storage and compute time (Neal, 1996; Welling & Teh, 2011).

To help overcome these problems, Balan et al. (2015) introduced an interesting model training method referred to as *Bayesian Dark Knowledge*. In the classification setting, Bayesian Dark Knowledge attempts to compress the Bayesian posterior predictive distribution induced by the full parameter posterior of a "teacher" network into a "student" network. The parameter posterior of the teacher network is represented through a Monte Carlo ensemble of specific instances of the teacher network (the teacher ensemble), and the analytically intractable posterior predictive distributions are approximated as Monte Carlo averages over the output of the networks in the teacher ensemble. The major advantage of this approach is that the computational complexity of prediction at test time is drastically reduced compared to computing Monte Carlo averages over a large ensemble of networks. As a result, methods of this type have the potential to be much better suited to learning models for deployment in resource constrained settings.

In this paper, we present a Bayesian posterior distillation framework that generalizes the Bayesian Dark Knowledge approach in several significant directions. The primary modeling and algorithmic contributions of this work are: (1) we generalize the target of distillation in the classification case from the posterior predictive distribution to general posterior expectations; (2) we generalize the

student architecture from being restricted to match the teacher architecture to being a free choice in the distillation procedure.

The primary empirical contributions of this work are (1) evaluating the distillation of both the posterior predictive distribution and expected posterior entropy across a range of models and data sets including manipulations of data sets that increase posterior uncertainty; and (2) evaluating the impact of the student model architecture on distillation performance including the investigation of sparsity-inducing regularization and pruning for student model architecture optimization. The key empirical findings are that (1) distilling into a student model that matches the architecture of the teacher, as in Balan et al. (2015), can be sub-optimal; and (2) student architecture optimization methods can identify significantly improved student models.

We note that the significance of generalizing distillation to arbitrary posterior expectations is that it allows us to capture a wider range of useful statistics of the posterior that are of interest from an uncertainty quantification perspective. As noted above, we focus on the case of distilling the expected posterior entropy in addition to the posterior predictive distribution itself. When combined with the entropy of the posterior predictive distribution, the expected posterior entropy enables disentangling model uncertainty (epistemic uncertainty) from fundamental uncertainty due to class overlap (aleatoric uncertainty). This distinction is extremely important in determining *why* predictions are uncertain for a given data case. Indeed, the difference between these two terms is the basis for the Bayesian active learning by disagreement (BALD) score used in active learning, which samples instances with the goal of minimizing model uncertainty (Houlsby et al., 2011).

The remainder of this paper is organized as follows. In the next section, we begin by presenting background material and related work in Section 2. In Section 3, we present the proposed framework and associated Generalized Posterior Expectation Distillation (GPED) algorithm. In Section 4, we present experiments and results. Additional details regarding data sets and experiments can be found in Appendix A, with supplemental results included in Appendix B.

## 2 BACKGROUND AND RELATED WORK

In this section we present background material on Bayesian inference for neural networks, and related work on approximate inference, and model compression and pruning.

### 2.1 BAYESIAN NEURAL NETWORKS

Let $p(y|\mathbf{x}, \theta)$ represent the probability distribution induced by a deep neural network classifier over classes $y \in \mathcal{Y} = \{1, .., C\}$ given feature vectors $\mathbf{x} \in \mathbb{R}^D$. The most common way to fit a model of this type given a data set $\mathcal{D} = \{(\mathbf{x}_i, y_i) | 1 \leq i \leq N\}$ is to use maximum conditional likelihood estimation, or equivalently, cross entropy loss minimization (or their penalized or regularized variants). However, when the volume of labeled data is low, there can be multiple advantages to considering a full Bayesian treatment of the model. Instead of attempting to find the single (locally) optimal parameter set $\theta_*$ according to a given criterion, Bayesian inference uses Bayes rule to define the posterior distribution $p(\theta|\mathcal{D}, \theta^0)$ over the unknown parameters $\theta$ given a prior distribution $P(\theta|\theta^0)$ with prior parameters $\theta^0$ as seen in Equation 1.

$$p(\theta|\mathcal{D}, \theta^0) = \frac{p(\mathcal{D}|\theta)p(\theta|\theta^0)}{\int p(\mathcal{D}|\theta)p(\theta|\theta^0)d\theta} \tag{1}$$

$$p(y|\mathbf{x}, \mathcal{D}, \theta^0) = \int p(y|\mathbf{x}, \theta)p(\theta|\mathcal{D}, \theta^0)d\theta = \mathbb{E}_{p(\theta|\mathcal{D}, \theta^0)}[p(y|\mathbf{x}, \theta)] \tag{2}$$

For prediction problems in machine learning, the quantity of interest is typically not the parameter posterior itself, but the posterior predictive distribution $p(y|\mathbf{x}, \mathcal{D}, \theta^0)$ obtained from it as seen in Equation 2. The primary problem with applying Bayesian inference to neural network models is that the distributions $p(\theta|\mathcal{D}, \theta^0)$ and $p(y|\mathbf{x}, \mathcal{D}, \theta^0)$ are not available in closed form, so approximations are required, which we discuss in the next section.

## 2.2 Approximate Inference Methods for Bayesian Neural Networks

Most Bayesian inference approximations studied in the machine learning literature are based on variational inference (VI) (Jordan et al., 1999) or Markov Chain Monte Carlo (MCMC) methods (Neal, 1996; Welling & Teh, 2011). In VI, an auxiliary distribution $q_\phi(\theta)$ is defined to approximate the true parameter posterior $p(\theta|\mathcal{D}, \theta^0)$. The variational parameters $\phi$ are selected to minimize the Kullback-Leibler (KL) divergence between $q_\phi(\theta)$ and $p(\theta|\mathcal{D}, \theta^0)$. Hinton & Van Camp (1993) first studied applying VI to neural networks. Graves (2011) later presented a method based on stochastic VI with improved scalability. In the closely related family of expectation propagation (EP) methods (Minka, 2001), Soudry et al. (2014) present an online EP algorithm for neural networks with the flexibility of representing both continuous and discrete weights. Hernández-Lobato & Adams (2015) present the probabilistic backpropagation (PBP) algorithm for approximate Bayesian learning of neural network models, which is an example of an assumed density filtering (ADF) algorithm that, like VI and EP, generally relies on simplified posterior densities.

The main drawback of VB, EP, and ADF is that they all typically result in biased posterior estimates for complex posterior distributions. MCMC methods provide an alternative family of sampling-based posterior approximations that are unbiased, but are often computationally more expensive to use at training time. MCMC methods allow for drawing a correlated sequence of samples $\theta_t \sim p(\theta|\mathcal{D}, \theta^0)$ from the parameter posterior. These samples can then be used to approximate the posterior predictive distribution as a Monte Carlo average as shown in Equation 3.

$$p(y|\mathbf{x}, \mathcal{D}, \theta^0) \approx \frac{1}{T} \sum_{t=1}^{T} p(y|\mathbf{x}, \theta_t); \quad \theta_t \sim p(\theta|\mathcal{D}, \theta^0) \tag{3}$$

Neal (1996) addressed the problem of Bayesian inference in neural networks using Hamiltonian Monte Carlo (HMC) to provide a set of posterior samples. A bottleneck with this method is that it uses the full dataset when computing the gradient needed by HMC, which is problematic for larger data sets. While this scalability problem has largely been solved by more recent methods such as stochastic gradient Langevin dynamics (SGLD) (Welling & Teh, 2011), the problem of needing to compute over a large set of samples when making predictions at test or deployment time remains.

Bayesian Dark Knowledge (Balan et al., 2015) is precisely aimed at reducing the test-time computational complexity of Monte Carlo-based approximations for neural networks. In particular, the method uses SGLD to approximate the posterior distribution using a set of posterior parameter samples. These samples can be thought of as an ensemble of neural network models with identical architectures, but different parameter values. This posterior ensemble is used as the "teacher" in a distillation process that trains a single "student" model to match the teacher ensemble's posterior predictive distribution (Hinton et al., 2015). The major advantage of this approach is that it can drastically reduce the test time computational complexity of posterior predictive inference relative to using a Monte Carlo average computed using many samples.

Finally, we note that with the advent of *Generative Adversarial Networks* (Goodfellow et al., 2014), there has also been work on generative models for approximating posterior sampling. Wang et al. (2018) and Henning et al. (2018) both propose methods for learning to generate samples that mimic those produced by SGLD. However, while these approaches may provide a speed-up relative to running SGLD itself, the resulting samples must still be used in a Monte Carlo average to compute a posterior predictive distribution in the case of Bayesian neural networks. This is again a potentially costly operation and is exactly the computation that Bayesian Dark Knowledge addresses.

## 2.3 Model Compression and Pruning

As noted above, the problem that Bayesian Dark Knowledge attempts to solve is reducing the test-time computational complexity of using a Monte-Carlo posterior to make predictions. In this work, we are particularly concerned with the issue of enabling test-time speed-storage-accuracy trade-offs. The relevant background material includes methods for network compression and pruning.

Previous work has shown that overparameterised deep learning models tend to show much better learnability. Further, it has also been shown that such overparameterised models rarely use their full capacity and can often be pruned back substatially without significant loss of generality (Hassibi et al., 1993; LeCun et al., 1989; Luo et al., 2017; Louizos et al., 2017; Frankle & Carbin, 2018;

Han et al., 2015; Zhang & Ou, 2018). Hassibi et al. (1993); LeCun et al. (1989) use the second order derivatives of the objective function to guide pruning network connections. More recently, Han et al. (2015) introduced a weight magnitude-based technique for pruning connections in deep neural networks using simple thresholding. Guo et al. (2016); Jin et al. (2016); Han et al. (2016) introduce thresholding methods which also support restoration of connections.

A related line of work includes pruning neurons/channels/filters instead of individual weights. Pruning these components explicitly reduces the number of computations by making the networks smaller. Group LASSO-based methods have the advantage of turning the pruning problem into a continuous optimization problem with a sparsity-inducing regularizer. Zhang & Ou (2018); Alvarez & Salzmann (2016); Wen et al. (2016); He et al. (2017) are some examples that use Group LASSO regularization at their core. Similarly Louizos et al. (2017) use hierarchical priors to prune neurons instead of weights. An advantage of these methods over ones which induce connection-based sparsity is that these methods directly produce smaller networks after pruning (e.g., fewer units or channels) as opposed to networks with sparse weight matrices. This makes it easier to realize the resulting computational savings, even on platforms that do not directly support sparse matrix operations.

## 3 PROPOSED FRAMEWORK

In this section, we describe our proposed framework for distilling general Bayesian posterior expectations for neural network classification models and discuss methods for enabling test-time speed-storage-accuracy trade-offs for flexible deployment of the resulting models.

### 3.1 GENERALIZED POSTERIOR EXPECTATIONS

There are many possible inferences of interest given a Bayesian parameter posterior $P(\theta|\mathcal{D}, \theta^0)$. We consider the general case of inferences that take the form of posterior expectations as shown in Equation 4 where $g(y, \mathbf{x}, \theta)$ is an arbitrary function of $y$, $\mathbf{x}$ and $\theta$.

$$E_{p(\theta|\mathcal{D},\theta^0)}[g(y, \mathbf{x}, \theta)] = \int g(y, \mathbf{x}, \theta)p(\theta|\mathcal{D}, \theta^0)d\theta \tag{4}$$

Important examples of functions $g(y, \mathbf{x}, \theta)$ include $g(y, \mathbf{x}, \theta) = p(y|\mathbf{x}, \theta)$, which results in a posterior expectation yielding the posterior predictive distribution $p(y|\mathbf{x}, \mathcal{D}, \theta^0)$, as used in Bayesian Dark Knowledge; $g(y, \mathbf{x}, \theta) = \sum_{y'=1}^{C} p(y'|\mathbf{x}, \theta) \log p(y'|\mathbf{x}, \theta)$, which yields the posterior predictive entropy $H(y|\mathbf{x}, \mathcal{D}, \theta^0)$[1]; and $g(y, \mathbf{x}, \theta) = p(y|\mathbf{x}, \theta)(1 - p(y|\mathbf{x}, \theta))$, which results in the posterior marginal variance $\sigma^2(y|\mathbf{x}, \mathcal{D}, \theta^0)$. While the posterior predictive distribution $p(y|\mathbf{x}, \mathcal{D}, \theta^0)$ is certainly the most important posterior inference from a predictive standpoint, the entropy and variance are also important from the perspective of uncertainty quantification.

### 3.2 GENERALIZED POSTERIOR EXPECTATION DISTILLATION

Our goal is to learn to approximate posterior expectations $E_{p(\theta|\mathcal{D},\theta^0)}[g(y, \mathbf{x}, \theta)]$ under a given teacher model architecture using a given student model architecture. The method that we propose takes as input the teacher model $p(y|\mathbf{x}, \theta)$, the prior $p(\theta|\theta^0)$, a labeled data set $\mathcal{D}$, an unlabeled data set $\mathcal{D}'$, the function $g(y, \mathbf{x}, \theta)$, a student model $f(y, \mathbf{x}|\phi)$, an online expectation estimator, and a loss function $\ell(\cdot, \cdot)$ that measures the error of the approximation given by the student model $f(y, \mathbf{x}|\phi)$. Similar to Balan et al. (2015), we propose an online distillation method based on the use of the SGLD sampler. We describe all of the components of the framework in the sections below, and provide a complete description of the resulting method in Algorithm 1.

**SGLD Sampler:** We define the prior distribution over the parameters $p(\theta|\theta^0)$ to be a spherical Gaussian distribution centered at $\mu = 0$ with precision $\tau$ (we thus have $\theta^0 = [\mu, \tau]$). We define $\mathcal{S}$ to be a minibatch of size M drawn from $\mathcal{D}$. $\theta_t$ denotes the parameter set sampled for the teacher model at sampling iteration $t$, while $\eta_t$ denotes the step size for the teacher model at iteration $t$. The Langevin noise is denoted by $z_t \sim \mathcal{N}(0, \eta_t I)$. The sampling update for SGLD is given by:

---

[1]Note that the posterior predictive entropy represents the average entropy integrated over the parameter posterior. It is not equal to the entropy of the posterior predictive distribution $p(y|\mathbf{x}, \mathcal{D}, \theta^0)$ in general.

$$\theta_{t+1} \leftarrow \theta_t + \frac{\eta_t}{2} \left( \nabla_\theta \log p(\theta|\theta^0) + \frac{N}{M} \sum_{i \in \mathcal{S}} \nabla_\theta \log p\left(y_i|x_i, \theta_t\right) \right) + z_t \tag{5}$$

**Distillation Procedure:** For the distillation learning procedure, we make use of a secondary unlabeled data set $\mathcal{D}' = \{\mathbf{x}_i | 1 \le i \le N'\}$. This data set could use feature vectors from the primary data set $\mathcal{D}$, or a larger data set. We note that due to autocorrelation in the sampled teacher model parameters $\theta_t$, we may not want to run a distillation update for every Monte Carlo sample drawn. We thus use two different iteration indices: $t$ for SGLD iterations and $s$ for distillation iterations.

On every distillation step $s$, we sample a minibatch $\mathcal{S}'$ from $\mathcal{D}'$ of size $M'$. For every data case $i$ in $\mathcal{S}'$, we update an estimate $\hat{g}_{yis}$ of the posterior expectation using the most recent parameter sample $\theta_t$, obtaining an updated estimate $\hat{g}_{yis+1} \approx E_{p(\theta|\mathcal{D},\theta^0)}[g(y, \mathbf{x}, \theta)]$ (we discuss update schemes in the next section). Next, we use the minibatch of examples $\mathcal{S}'$ to update the student model. To do so, we take a step in the gradient direction of the regularized empirical risk of the student model as shown below where $\alpha_s$ is the student model learning rate, $R(\phi)$ is the regularizer, and $\lambda$ is the regularization hyper-parameter. We next discuss the estimation of the expectation targets $\hat{g}_{yis}$.

$$\phi_{s+1} \leftarrow \phi_s + \alpha_s \left( \frac{N'}{M'} \sum_{i \in \mathcal{S}'} \sum_{y \in \mathcal{Y}} \nabla_\phi \ell\left(\hat{g}_{yis+1}, f(y, \mathbf{x}_i|\phi_s)\right) + \lambda \nabla_\phi R(\phi_s) \right) \tag{6}$$

**Expectation Estimation:** Given an explicit collection of posterior samples $\theta_1, ..., \theta_s$, the standard Monte Carlo estimate of $E_{p(\theta|\mathcal{D},\theta^0)}[g(y, \mathbf{x}, \theta)]$ is simply $\hat{g}_{yis} = \frac{1}{s} \sum_{j=1}^{s} g(y, \mathbf{x}_i, \theta_j)$. However, this estimator requires retaining the sequence of samples $\theta_1, ..., \theta_s$, which may not be feasible in terms of storage cost. Instead, we consider the application of an online update function. We define $m_{is}$ to be the count of the number of times data case $i$ has been sampled up to and including distillation iteration $s$. An online update function $U(\hat{g}_{yis}, \theta_t, m_{is})$ takes as input the current estimate of the expectation, the current sample of the model parameters, and the number of times data case $i$ has been sampled, and produces an updated estimate of the expectation $\hat{g}_{yis+1}$. Below, we define two different versions of the function. $U_s(\hat{g}_{yis}, \theta_t, m_{is})$, updates $\hat{g}_{yis}$ using the current sample only, while $U_o(\hat{g}_{yis}, \theta_t, m_{is})$ performs an online update equivalent to a full Monte Carlo average.

$$U_s(\hat{g}_{yis}, \theta_t, m_{is}) = g(y, \mathbf{x}_i, \theta_t) \tag{7}$$

$$U_o(\hat{g}_{yis}, \theta_t, m_{is}) = \frac{1}{m_{is+1}} \left(m_{is} \cdot \hat{g}_{yis} + g(y, \mathbf{x}_i, \theta_t)\right) \tag{8}$$

We note that both update functions provide unbiased estimates of $E_{p(\theta|\mathcal{D},\theta^0)}[g(y, \mathbf{x}, \theta)]$ after a suitable burn-in time $B$. The online update $U_o()$ will generally result in much lower variance in the estimated values of $\hat{g}_{yis}$, but it comes at the cost of needing to explicitly maintain the expectation estimates $\hat{g}_{yis}$ across learning iterations, increasing the storage cost of the algorithm. It is worthwhile noting that the extra storage and computation cost required by $U_o$ grows linearly in the size of the training set for the student. By contrast, the fully stochastic update is memoryless in terms of past expectation estimates, so the estimated expectations $\hat{g}_{yis}$ do not need to be retained across iterations resulting in a space savings.

**General Algorithm and Special Cases:** We show a complete description of the proposed method in Algorithm 1. The algorithm takes as input the teacher model $p(y|\mathbf{x}, \theta)$, the parameters of the prior $P(\theta|\theta^0)$, a labeled data set $\mathcal{D}$, an unlabeled data set $\mathcal{D}'$, the function $g(y, \mathbf{x}, \theta)$, the student model $f(y, \mathbf{x}|\phi)$, an online expectation estimator $U(\hat{g}_{yis}, \theta_t, m_{is})$, a loss function $\ell(\cdot, \cdot)$ that measures the error of the approximation given by $f(y, \mathbf{x}|\phi)$, a regularization function $R()$ and regularization hyper-parameter $\lambda$, minibatch sizes $M$ and $M'$, the thinning interval parameter $H$, the SGLD burn-in time parameter $B$ and step size schedules for the step sizes $\eta_t$ and $\alpha_s$.

We note that the original Bayesian Dark Knowledge method is recoverable as a special case of this framework via the the choices $g(y, \mathbf{x}, \theta) = p(y|\mathbf{x}, \theta)$, $\ell(p, q) = -p \log(q)$, $U = U_s$ and $p(y|\mathbf{x}, \theta) = f(y, \mathbf{x}, \phi)$ (e.g., the architecture of the student is selected to match that of the teacher). The original approach also uses a distillation data set $\mathcal{D}'$ obtained from $\mathcal{D}$ by adding randomly

---

**Algorithm 1** Generalized Posterior Expectation Distillation

1: **procedure** GPED($\mathcal{D}, \mathcal{D}', p(y|\mathbf{x}, \theta), \theta^0, g, f, U, \ell, R, M, M', H, B, \lambda, \{\eta_t\}_{t=1}^T, \{\alpha_s\}_{s=1}^S$ )
2:     Initialize $s = 0$, $\phi_0$, $\theta_0$, $\hat{g}_{yi0} = 0$, $m_{i0} = 0$, $\eta_0$
3:     **for** $t = 0$ to $T$ **do**
4:         Sample $\mathcal{S}$ from $\mathcal{D}$ with $|\mathcal{S}| = M$
5:         $\theta_{t+1} \leftarrow \theta_t + \frac{\eta_t}{2}\left(\nabla_\theta \log p(\theta|\theta^0) + \frac{N}{M}\sum_{i\in\mathcal{S}}\nabla_\theta \log p\left(y_i|x_i, \theta_t\right)\right) + z_t$
6:         **if**  $\mod(t, H) = 0$ and $t > B$ **then**
7:             Sample $\mathcal{S}'$ from $\mathcal{D}'$ with $|\mathcal{S}'| = M'$
8:             **for** $i \in \mathcal{S}'$ **do**
9:                 $\hat{g}_{yis+1} \leftarrow U(\hat{g}_{yis}, \theta_t, m_{is})$
10:                $m_{is+1} \leftarrow m_{is} + 1$
11:            **end for**
12:            $\phi_{s+1} \leftarrow \phi_s + \alpha_s \left(\frac{N'}{M'}\sum_{i\in\mathcal{S}'}\sum_{y\in\mathcal{Y}}\nabla_\phi \ell\left(\hat{g}_{yis+1}, f(y, \mathbf{x}_i|\phi_s)\right) + \lambda\nabla_\phi R(\phi_s)\right)$
13:            $s \leftarrow s + 1$
14:        **end if**
15:    **end for**
16: **end procedure**

---

generated noise to instances from $\mathcal{D}$ on each distillation iteration, taking advantage of the fact that the choice $U = U_s$ means that no aspect of the algorithm scales with $|\mathcal{D}'|$.

Our general framework allows for other trade-offs, including reducing the variance in the estimates of $\hat{g}_{yis}$ at the cost of additional storage in proportion to $|\mathcal{D}'|$. We also note that note that the loss function $\ell(p, q) = -p\log(q)$ and the choice $g(y, \mathbf{x}, \theta) = p(y|\mathbf{x}, \theta)$ are somewhat of a special case when used together as even when the full stochastic expectation update $U_s$ is used, the resulting distillation parameter gradient is unbiased. To distill posterior entropy, we set $g(y, \mathbf{x}, \theta) = \sum_{y\in\mathcal{Y}} p(y|\mathbf{x}, \theta)\log p(y|\mathbf{x}, \theta)$, $U = U_o$ and $\ell(h, h') = |h - h'|$.

### 3.3 MODEL COMPRESSION AND PRUNING

One of the primary motivations for the original Bayesian Dark Knowledge approach is that it provides an approximate inference framework that results in significant computational and storage savings at test time. However, a drawback of the original approach is that the architecture of the student is chosen to match that of the teacher. As we will show in Section 4, this will sometimes result in a student network that has too little capacity to represent a particular posterior expectation accurately. On the other hand, if we plan to deploy the student model in a low resource compute environment, the teacher architecture may not meet the specified computational constraints. In either case, we need a general approach for selecting an architecture for the student model.

To begin to explore this problem, we consider to basic approaches to choosing student model architectures that enable trading off test time inference speed and storage for accuracy. A helpful aspect of the distillation process relative to a de novo architecture search problem is that the architecture of the teacher model is available as a starting point. As a first approach, we consider wrapping the proposed GPED algorithm with an explicit search over a set of student models that are "close" to the teacher. Specifically, we consider a search space obtained by starting from the teacher model and applying a width multiplier to the width of every fully connected layer and a kernel multiplier to the number of kernels in every convolutional layer. While this search requires exponential time in the number of layers, it provides a baseline for evaluating other methods.

As an alternative approach with better computational complexity, we leverage the regularization function $R(\phi)$ included in the GPED framework to prune a large initial network using group $\ell_1/\ell_2$ regularization (Zhang & Ou, 2018; Wen et al., 2016). To apply this approach, we first must partition the parameters in the parameter vector $\phi$ across $K$ groups $\mathcal{G}_k$. The form of the regularizer is $R(\phi) = \sum_{k=1}^K \left(\sum_{j\in\mathcal{G}_k} \phi_j^2\right)^{1/2}$. As is well-established in the literature, this regularizer causes all parameters in a group to go to zero simultaneously when they are not needed in a model. To use it for model pruning for a unit in a fully connected layer, we collect all of that unit's inputs into a group. Similarly, we collect all of the incoming weights for a particular channel in a convolution layer

together into a group. If all incoming weights associated with a unit or a channel have magnitude below a small threshold $\epsilon$, we can explicitly remove them from the model, obtaining a more compact architecture. We also fine-tune our models after pruning.

Finally, we note that any number of weight compressing, pruning, and architecture search methods could be combined with the GPED framework. Our goal is not to exhaustively compare such methods, but rather to demonstrate that GPED is sensitive to the choice of student model to highlight the need for additional research on the problem of selecting student model architectures.

# 4 EXPERIMENTS AND RESULTS

In this section, we present experiments and results evaluating the proposed approach using multiple data sets, posterior expectations, teacher model architectures, student model architectures and basic architecture search methods. We begin by providing an overview of the experimental protocols used.

## 4.1 EXPERIMENTAL PROTOCOLS

**Data Sets:** We use the MNIST (LeCun, 1998) and CIFAR10 (Krizhevsky et al., 2009) data sets as base data sets in our experiments. In the case of MNIST, posterior predictive uncertainty is very low, so we introduce two different modifications to explore the impact of uncertainty on distillation performance. The first modification is simply to subsample the data. The second modification is to introduce occlusions into the data set using randomly positioned square masks of different sizes, resulting in masking rates from $0\%$ to $86.2\%$. For CIFAR10, we only use sub-sampling. Full details for both data sets and the manipulations applied can be found in Appendix A.1.

**Models:** We evaluate a total of three teacher models in this work: a three-layer fully connected network (FCNN) for MNIST matching the architecture used by Balan et al. (2015), a four-layer convolutional network for MNIST, and a five-layer convolutional network for CIFAR10. Full details of the teacher model architectures are given in Appendix A.2. For exhaustive search for student model architectures, we use the teacher model architectures as base models and search over a space of layer width multipliers $K_1$ and $K_2$ that can be used to expand sets of layers in the teacher models. A full description of the search space of student models can be found in Appendix A.2.

**Distillation Procedures:** We consider distilling both the posterior predictive distribution and the posterior entropy, as described in the previous section. For the posterior predictive distribution, we use the stochastic expectation estimator $U_s$ while for entropy we used the full online update $U_o$. We allow $B = 1000$ burn-in iterations and total of $T = 10^6$ training iterations. The prior hyper-parameters, learning rate schedules and other parameters vary by data set or distillation target and are fully described in Appendix A.2.

## 4.2 EXPERIMENT 1: DISTILLING POSTERIOR EXPECTATIONS

For this experiment, we use the MNIST and CIFAR10 datasets without any subsampling or masking. For each dataset and model, we consider separately distilling the posterior predictive distribution and the posterior entropy. We fix the architecture of the student to match that of the teacher. To evaluate the performance while distilling the posterior predictive distribution, we use the negative log-likelihood (NLL) of the model on the test set. For evaluating the performance of distilling posterior entropy, we use the mean absolute difference between the teacher ensemble's entropy estimate and the student model output on the test set. The results are given in Table 1. First, we note that the FCNN NLL results on MNIST closely replicate the results in Balan et al. (2015), as expected. We also note that the error in the entropy is low for both the FCNN and CNN architectures on MNIST. However, the student model fails to match the NLL of the teacher on CIFAR10 and the entropy MAE is also relatively high. In Experiment 2, we will investigate the effect of increasing uncertainty on models applied to both data sets, while in Experiment 3 we will search for student model architectures that improve performance.

| Model & Dataset | Teacher NLL | Student NLL | MAE (Entropy) |
|---|---|---|---|
| FCNN - MNIST | 0.052 | 0.082 | 0.016 |
| CNN - MNIST | 0.022 | 0.053 | 0.016 |
| CNN - CIFAR10 | 0.671 | 0.932 | 0.245 |

Table 1: Results of posterior distillation when the student architecture is fixed to match the teacher architecture and the base data sets are used with no sub-sampling or occlusion.

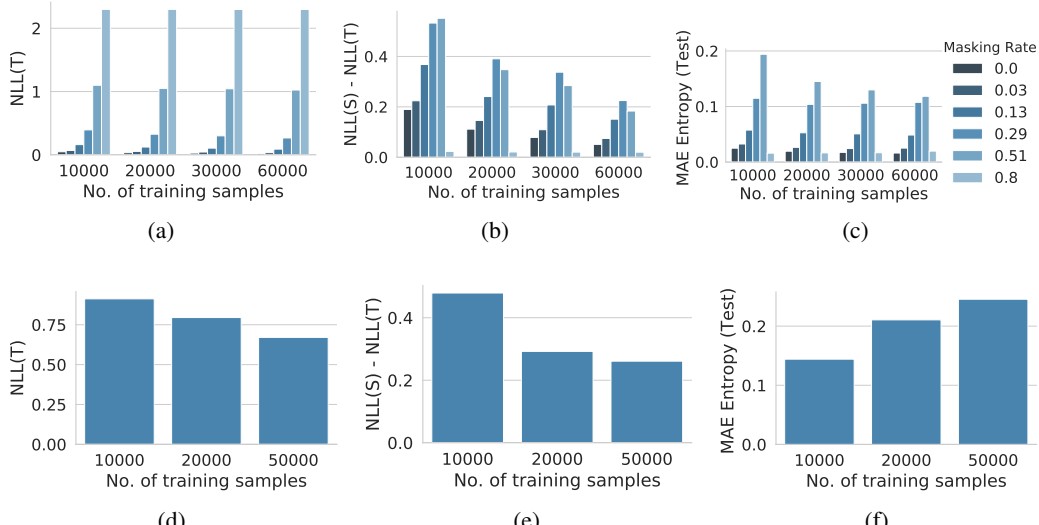

Figure 1: Top Row: Distillation performance using CNNs on MNIST while varying data set size and masking rate. (a) Test negative log likelihood of the teacher posterior predictive distribution. (b) Difference in test negative log likelihood between student and teacher posterior predictive distribution estimates. (c) Difference between teacher and student posterior entropy estimates on test data set. Bottom Row: Distillation performance using CNNs on CIFAR10 while varying data set size. (d) Test negative log likelihood of the teacher posterior predictive distribution. (e) Difference in test negative log likelihood between student and teacher posterior predictive distribution estimates. (f) Difference between teacher and student posterior entropy estimates on test data set. In the plots above, S denotes the student and T denotes the teacher.

## 4.3 EXPERIMENT 2: ROBUSTNESS TO UNCERTAINTY

This experiment builds on Experiment 1 by exploring methods for increasing posterior uncertainty on MNIST (sub-sampling and masking) and CIFAR10 (sub-sampling). We consider the cross product of four sub-sampling rates and six masking rates for MNIST and three sub-sampling rates for CIFAR10. We consider the posterior predictive distribution and posterior entropy distillation targets. For the posterior predictive distribution we report the negative log likelihood (NLL) of the teacher, and the NLL gap between the teacher and student. For entropy, we report the mean absolute error between the teacher ensemble and the student. All metrics are evaluated on held-out test data. We also restrict the experiment to the case where the student architecture matches the teacher architecture, mirroring the Bayesian Dark Knowledge approach. In Figure 1, we show the results for the convolutional models on MNIST and CIFAR10 respectively. The FCNN results are similar to the CNN results on MNIST and are shown in Figure 4 in Appendix B. In Appendix B, we also provide a performance comparison between the $U_o$ and $U_s$ estimators while distilling posterior expectations.

As expected, the the NLL of the teacher decreases as the data set size decreases. We observe that changing the number of training samples has a similar effect on NLL gap for both CIFAR10 and MNIST. More specifically, for any fixed masking rate of MNIST (and zero masking rate for CIFAR10), we can see that the NLL difference between the student and teacher decreases with increasing training data. However, for MNIST we can see that the teacher NLL increases much more rapidly as a function of the masking rate. Moreover, the gap between the teacher and student peaks

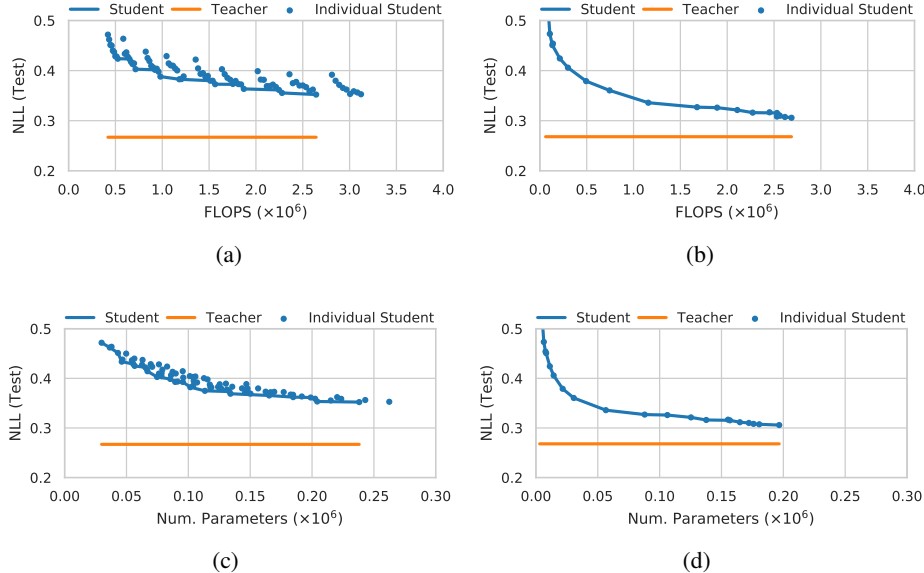

Figure 2: NLL-Storage-Computation tradeoff while using CNNs on MNIST with masking rate 29%. (a,b) Test negative log likelihood of posterior predictive distribution vs FLOPS found using exhaustive search and group $\ell_1/\ell_2$ with pruning. (c,d) Test negative log likelihood of posterior predictive distribution vs storage found using exhaustive search and group $\ell_1/\ell_2$ with pruning. The optimal student model for this configuration is obtained with group $\ell_1/\ell_2$ pruning. It has approximately $6.6\times$ the number of parameters and $6.4\times$ the FLOPS of the base student model.

for moderate values of the masking rate. This fact is explained through the observation that when the masking rate is low, posterior uncertainty is low, and distillation is relatively easy. On the other hand, when the masking rate is high, the teacher essentially outputs the uniform distribution for every example, which is very easy for the student to represent. As a result, the moderate values of the masking rate result in the hardest distillation problem and thus the largest performance gap. For varying masking rates, we see exactly the same trend for the gap in posterior entropy predictions on MNIST. However, the gap for entropy prediction increases as a function of data set size for CIFAR10. Finally, as we would expect, the performance of distillation using the $U_o$ estimator is almost always better than that of the $U_s$ estimator (refer Appendix B).

The key finding of this experiment is simply that the quality of the approximations provided by the student model varies as a function of properties of the underlying data set. Indeed, restricting the student architecture to match the teacher can sometimes result in significant performance gaps. In the next experiment, we address the problem of searching for improved student model architectures.

## 4.4 EXPERIMENT 3: TOWARDS STUDENT MODEL ARCHITECTURE SEARCH

In this experiment, we compare the exhaustive search to the group $\ell_1/\ell_2$ (group lasso) regularizer combined with pruning. For the pruning approach, we start with the largest student model considered under exhaustive search, and prune back from there using different regularization parameters $\lambda$, leading to different student model architectures. We present results in terms of performance versus computation time (estimated in FLOPS), as well as performance vs storage cost (estimated in number of parameters). As performance measures for the posterior predictive distribution, we consider accuracy and negative log likelihood. For entropy, we use mean absolute error. In all cases results are reported on test data. We consider both fully connected and convolutional models.

Figure 2 shows results for negative the log likelihood (NLL) of the convolutional model on MNIST with masking rate 29% and 60,000 training samples. We select this setting as illustrative of a difficult case for posterior predictive distribution distillation. We plot NLL vs FLOPS and NLL vs storage for all points encountered in each search. The solid blue line indicates the Pareto frontier.

First, we note that the baseline student model (with architecture matching the teacher) from Experiment 2 on MNIST achieves an NLL of $0.469$ at approximately $0.48 \times 10^6$ FLOPs and $0.03 \times 10^6$ parameters on this configuration of the data set. We can see that both methods for selecting student architectures provide a highly significant improvement over the baseline student architectures. On MNIST, the NLL is reduced to $0.30$. Further, we can also see that the group $\ell_1/\ell_2$ approach is able to obtain much better NLL at the same computation and storage cost relative to the exhaustive search method. Lastly, the group $\ell_1/\ell_2$ method is able to obtain models on MNIST at less than $50\%$ the computational cost needed by the baseline model with only a small loss in performance. Results for other models and distillation targets show similar trends and are presented in Appendix B. Additional experimental details are given in Appendix A.2.

In summary, the key finding of this experiment is that the capacity of the student model has a significant impact on the performance of the distillation procedure, and methods for optimizing the student architecture are needed to achieve a desired speed-storage-accuracy trade-off.

## 5    Conclusions & Future Directions

We have presented a framework for distilling expectations with respect to the Bayesian posterior distribution of a deep neural network that generalizes the Bayesian Dark Knowledge approach in several significant directions. Our results show that the performance of posterior distillation can be highly sensitive to the architecture of the student model, but that basic architecture search methods can help to identify student model architectures with improved speed-storage-accuracy trade-offs. There are many directions for future work including considering the distillation of a broader class of posterior statistics including percentiles, assessing and developing more advanced student model architecture search methods, and applying the framework to larger state-of-the-art models.

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

## A   DATASETS AND MODEL DETAILS

### A.1   DATASETS

As noted earlier in the paper, the original empirical investigation of Bayesian Dark Knowledge for classification focused on the MNIST data set (LeCun, 1998). However, the models fit to the MNIST data set have very low posterior uncertainty and we argue that it is thus a poor benchmark for assessing the performance of posterior distillation methods. In this section, we investigate two orthogonal modifications of the standard MNIST data set to increase uncertainty: reducing the training set size and masking regions of the input images. Our goal is to produce a range of benchmark problems with varying posterior predictive uncertainty. We also use the CIFAR10 data set (Krizhevsky et al., 2009) in our experiments and employ the same subsampling technique.

**MNIST:** The full MNIST dataset consists of 60,000 training images and 10,000 test images, each of size $28 \times 28$, distributed among 10 classes LeCun (1998). As a first manipulation, we consider sub-sampling the labeled training data to include 10,000, 20,000, 30,000 or all 60,000 data cases in the primary data set $\mathcal{D}$ when performing posterior sampling for the teacher model. Importantly, we use all 60,000 unlabeled training cases in the distillation data set $\mathcal{D}'$. This allows us de-couple the impact of reduced labeled training data on posterior predictive distributions from the effect of the amount of unlabeled data available for distillation.

As a second manipulation, we generate images with occlusions by randomly masking out parts of each available training and test image. For generating such images, we randomly choose a square $m \times m$ region (mask) and set the value for pixels in that region to 0. Thus, the masking rate for a $28 \times 28$ MNIST image corresponding to the mask of size $m \times m$ is given by $r = \frac{m \times m}{28 \times 28}$. We illustrate original and masked data in Figure 3. We consider a range of square masks resulting in masking rates between 0% and 86.2%.

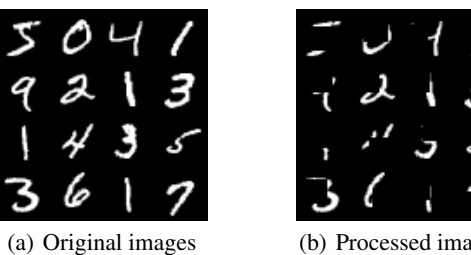

(a) Original images          (b) Processed images

Figure 3: Example MNIST data after masking with $m = 14$.

**CIFAR10:** The full CIFAR10 dataset consists of 50,000 training images and 10,000 test images, each of size $32 \times 32$ pixels. We sub-sample the data into a primary training sets $\mathcal{D}$ containing 10,000,

20,000, and 50,000 images. As with MNIST, the sub-sampling is limited to training the teacher model only and we utilize all the 50,000 unlabeled training images in the distillation data set $\mathcal{D}'$.

## A.2 MODELS

To demonstrate the generalizability of our methods to a range of model architectures, we run our experiments with both fully-connected, and convolutional neural networks. We note that our goal in this work is not to evaluate the GPED framework on state-of-the-art architectures, but rather to provide illustrative results and establish methodology for assessing the impact of several factors including the level of uncertainty and the architecture of the student model.

**Teacher Models:** We begin by defining the architectures used for the teacher model as follows:

1. **FCNN (MNIST)**: We use a 3-layer fully connected neural network. The architecture used is: Input(784)-FC(400)-FC(400)-FC(output). This matches the architecture used by Balan et al. (2015).

2. **CNN (MNIST)**: For a CNN, we use two consecutive sets of 2D convolution and max-pooling layers, followed by two fully-connected layers. The architecture used is: Input(1, (28,28))-Conv(num_kernels=10, kernel_size=4, stride=1) - MaxPool(kernel_size=2) - Conv(num_kernels=20, kernel_size=4, stride=1) - MaxPool(kernel_size=2) - FC (80) - FC (output).

3. **CNN (CIFAR10)**: Similar to the CNN architecture used for MNIST, we use two consecutive sets of 2D convolution and max-pooling layers followed by fully-connected layers. Conv(num_kernels=16, kernel_size=5) - MaxPool(kernel_size=2) - Conv(num_kernels=32, kernel_size=5) - MaxPool(kernel_size=2) - FC(200) - FC (50) - FC (output).

In the architectures mentioned above, the "output" size will change depending on the expectation that we're distilling. For classification, the output size will be 10 for both datasets, while for the case of entropy, it will be 1. We use *ReLU* non-linearities everywhere between the hidden layers. For the final output layer, *softmax* is used for classification. In the case of entropy, we use an exponential activiation to ensure positivity.

**Student Models:** The student models used in our experiments use the above mentioned architectures as the base architecture. For explicitly searching the space of the student models, we use a set of width multipliers starting from the teacher architecture. The space of student architectures corresponding to each teacher model defined earlier is given below. The width multiplier values of $K_1$ and $K_2$ are determined differently for each of the experiments, and thus will be mentioned in later sections.

1. **FCNN (MNIST)**: Input(784)-FC($400 \cdot K_1$)-FC($400 \cdot K_2$)-FC(output).

2. **CNN (MNIST)**: Input(1, (28,28))-Conv(num_kernels=$\lfloor 10 \cdot K_1 \rfloor$, kernel_size=4, stride=1) - MaxPool(kernel_size=2) - Conv(num_kernels=$\lfloor 20 \cdot K_1 \rfloor$, kernel_size=4, stride=1) - MaxPool(kernel_size=2) - FC ($\lfloor 80 \cdot K_2 \rfloor$) - FC (output).

3. **CNN (CIFAR10)**: Input(3, (32,32))-Conv(num_kernels=$\lfloor 16 \cdot K_1 \rfloor$, kernel_size=5) - MaxPool(kernel_size=2) - Conv(num_kernels=$\lfloor 16 \cdot K_1 \rfloor$, kernel_size=5) - MaxPool(kernel_size=2) - FC ($\lfloor 200 \cdot K_2 \rfloor$) - FC ($\lfloor 50 \cdot K_2 \rfloor$) - FC (output).

**Model and Distillation Hyper-Parameters**: We run the distillation procedure using the following hyperparameters: fixed teacher learning rate $\eta_t = 4 \times 10^{-6}$ for models on MNIST and $\eta_t = 2 \times 10^{-6}$ for models on CIFAR10, teacher prior precision $\tau = 10$, initial student learning rate $\alpha_s = 10^{-3}$, student dropout rate $p = 0.5$ for fully-connected models on MNIST (and zero otherwise), burn-in iterations $B = 1000$, thinning interval $H = 100$ for distilling predictive means and $H = 10$ for distilling entropy values, and total training iterations $T = 10^6$. For training the student model, we use the *Adam* algorithm (instead of plain steepest descent as indicated in Algorithm 1) and set a learning schedule for the student such that it halves its learning rate every 200 epochs for models on MNIST, and every 400 epochs for models on CIFAR10. Also, note that we only apply the regularization function $R(\phi_s)$ while doing Group $\ell_1/\ell_2$ pruning. Otherwise, we use dropout as indicated before.

**Hyper-parameters for Group $\ell_1/\ell_2$ pruning experiments**: For experiments involving group $\ell_1/\ell_2$ regularizer, the regularization strength values $\lambda$ are chosen from a log-scale ranging from $10^{-8} to 10 - 3$.

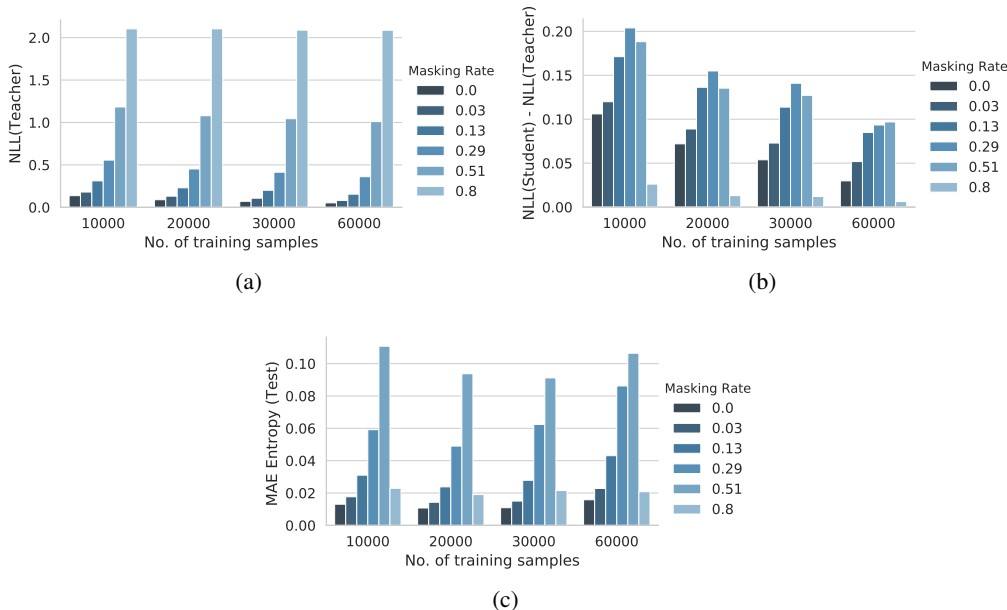

Figure 4: Distillation performance using Fully-Connected Networks on MNIST while varying data set size and masking rate. (a) Test negative log likelihood of the teacher posterior predictive distribution. (b) Difference in test negative log likelihood between teacher and student posterior predictive distribution estimates. (c) Difference between teacher and student posterior entropy estimates on test data set.

When using Group $\ell_1/\ell_2$ regularizer, we do not use dropout for the student model. The number of fine-tuning epochs for models on MNIST and CIFAR100 are 600 and 800 respectively. At the start of fine-tuning, we also reinitialize the student learning rate $\alpha_t = 10^{-4}$ for fully-connected models and $\alpha_t = 10^{-3}$ for convolutional models. The magnitude threshold for pruning is $\epsilon = 10^{-3}$.

## B  SUPPLEMENTAL EXPERIMENTS AND RESULTS

**Supplemental Results for Experiment 2: Robustness to Uncertainty** In Figure 4, we demonstrate the results of Experiment 2 (Section 4.3), on fully-connected networks for MNIST. Additionally, in Tables [2-4], we provide a performance comparison between $U_o$ and $U_s$ estimators while distilling posterior expectations for all model-data set combinations. We follow the same experimental configurations as in Experiment 2.

**Supplemental Results for Experiment 3: Towards Student Model Architecture Search** The additional results from running Experiment 3 (Section 4.4) on different combinations of model type, dataset, and performance metrics have been given in Figures[5 - 12].

| Num. training samples | Masking rate | NLL (Teacher) | NLL (Student, $U_o$) | NLL (Student, $U_s$) | MAE (Entropy, $U_o$) | MAE (Entropy, $U_s$) |
|---|---|---|---|---|---|---|
| 10000 | 0 | 0.048 | 0.214 | 0.218 | 0.025 | 0.030 |
| | 0.03 | 0.069 | 0.274 | 0.274 | 0.033 | 0.038 |
| | 0.13 | 0.161 | 0.509 | 0.509 | 0.058 | 0.069 |
| | 0.29 | 0.394 | 0.902 | 0.907 | 0.115 | 0.129 |
| | 0.51 | 1.099 | 1.615 | 1.630 | 0.194 | 0.170 |
| | 0.8 | 2.298 | 2.301 | 2.301 | 0.016 | 0.019 |
| 20000 | 0 | 0.034 | 0.126 | 0.126 | 0.020 | 0.021 |
| | 0.03 | 0.054 | 0.180 | 0.181 | 0.026 | 0.030 |
| | 0.13 | 0.123 | 0.342 | 0.344 | 0.053 | 0.066 |
| | 0.29 | 0.326 | 0.684 | 0.697 | 0.104 | 0.122 |
| | 0.51 | 1.050 | 1.369 | 1.378 | 0.145 | 0.150 |
| | 0.8 | 2.298 | 2.300 | 2.299 | 0.016 | 0.020 |
| 30000 | 0 | 0.028 | 0.084 | 0.086 | 0.017 | 0.019 |
| | 0.03 | 0.044 | 0.132 | 0.134 | 0.024 | 0.027 |
| | 0.13 | 0.106 | 0.292 | 0.294 | 0.051 | 0.061 |
| | 0.29 | 0.300 | 0.620 | 0.618 | 0.106 | 0.120 |
| | 0.51 | 1.044 | 1.307 | 1.308 | 0.130 | 0.141 |
| | 0.8 | 2.296 | 2.297 | 2.296 | 0.017 | 0.021 |
| 60000 | 0 | 0.022 | 0.053 | 0.053 | 0.016 | 0.017 |
| | 0.03 | 0.035 | 0.088 | 0.090 | 0.025 | 0.026 |
| | 0.13 | 0.090 | 0.219 | 0.221 | 0.049 | 0.058 |
| | 0.29 | 0.267 | 0.463 | 0.472 | 0.108 | 0.120 |
| | 0.51 | 1.024 | 1.184 | 1.187 | 0.118 | 0.127 |
| | 0.8 | 2.297 | 2.297 | 2.297 | 0.020 | 0.023 |

Table 2: Performance comparison between $U_o$ and $U_s$ estimators for convolutional neural network on MNIST. The NLL results correspond to the case of distilling the posterior predictive distribution while the MAE on entropy results correspond to the case of distilling the expectation of predictive entropy.

| Num. training samples | Masking rate | NLL (Teacher) | NLL (Student, $U_o$) | NLL (Student, $U_s$) | MAE (Entropy, $U_o$) | MAE (Entropy, $U_s$) |
|---|---|---|---|---|---|---|
| 10000 | 0 | 0.137 | 0.184 | 0.243 | 0.013 | 0.018 |
| | 0.03 | 0.180 | 0.233 | 0.300 | 0.018 | 0.023 |
| | 0.13 | 0.312 | 0.389 | 0.483 | 0.031 | 0.040 |
| | 0.29 | 0.556 | 0.637 | 0.760 | 0.059 | 0.089 |
| | 0.51 | 1.183 | 1.229 | 1.371 | 0.111 | 0.135 |
| | 0.8 | 2.103 | 2.111 | 2.129 | 0.023 | 0.019 |
| 20000 | 0 | 0.089 | 0.115 | 0.161 | 0.011 | 0.015 |
| | 0.03 | 0.131 | 0.165 | 0.220 | 0.014 | 0.021 |
| | 0.13 | 0.230 | 0.280 | 0.366 | 0.024 | 0.042 |
| | 0.29 | 0.452 | 0.510 | 0.607 | 0.049 | 0.104 |
| | 0.51 | 1.080 | 1.120 | 1.215 | 0.094 | 0.112 |
| | 0.8 | 2.104 | 2.108 | 2.117 | 0.019 | 0.021 |
| 30000 | 0 | 0.071 | 0.083 | 0.124 | 0.011 | 0.014 |
| | 0.03 | 0.107 | 0.129 | 0.180 | 0.015 | 0.021 |
| | 0.13 | 0.201 | 0.243 | 0.314 | 0.028 | 0.052 |
| | 0.29 | 0.414 | 0.459 | 0.555 | 0.062 | 0.105 |
| | 0.51 | 1.044 | 1.082 | 1.172 | 0.091 | 0.105 |
| | 0.8 | 2.089 | 2.092 | 2.101 | 0.022 | 0.023 |
| 60000 | 0 | 0.052 | 0.054 | 0.082 | 0.016 | 0.020 |
| | 0.03 | 0.081 | 0.094 | 0.133 | 0.023 | 0.034 |
| | 0.13 | 0.155 | 0.186 | 0.240 | 0.043 | 0.068 |
| | 0.29 | 0.360 | 0.398 | 0.471 | 0.086 | 0.109 |
| | 0.51 | 1.010 | 1.033 | 1.107 | 0.106 | 0.099 |
| | 0.8 | 2.088 | 2.089 | 2.094 | 0.021 | 0.022 |

Table 3: Performance comparison between $U_o$ and $U_s$ estimators for fully-connected network on MNIST. The NLL results correspond to the case of distilling the posterior predictive distribution while the MAE on entropy results correspond to the case of distilling the expectation of predictive entropy.

| Num. training samples | NLL (Teacher) | NLL (Student, $U_o$) | NLL (Student, $U_s$) | MAE (Entropy, $U_o$) | MAE (Entropy, $U_s$) |
|---|---|---|---|---|---|
| 10000 | 0.912 | 1.372 | 1.391 | 0.144 | 0.192 |
| 20000 | 0.798 | 1.184 | 1.179 | 0.210 | 0.231 |
| 50000 | 0.671 | 0.924 | 0.932 | 0.245 | 0.290 |

Table 4: Performance comparison between $U_o$ and $U_s$ estimators for convolutional neural network on CIFAR10. The NLL results correspond to the case of distilling the posterior predictive distribution while the MAE on entropy results correspond to the case of distilling the expectation of predictive entropy.

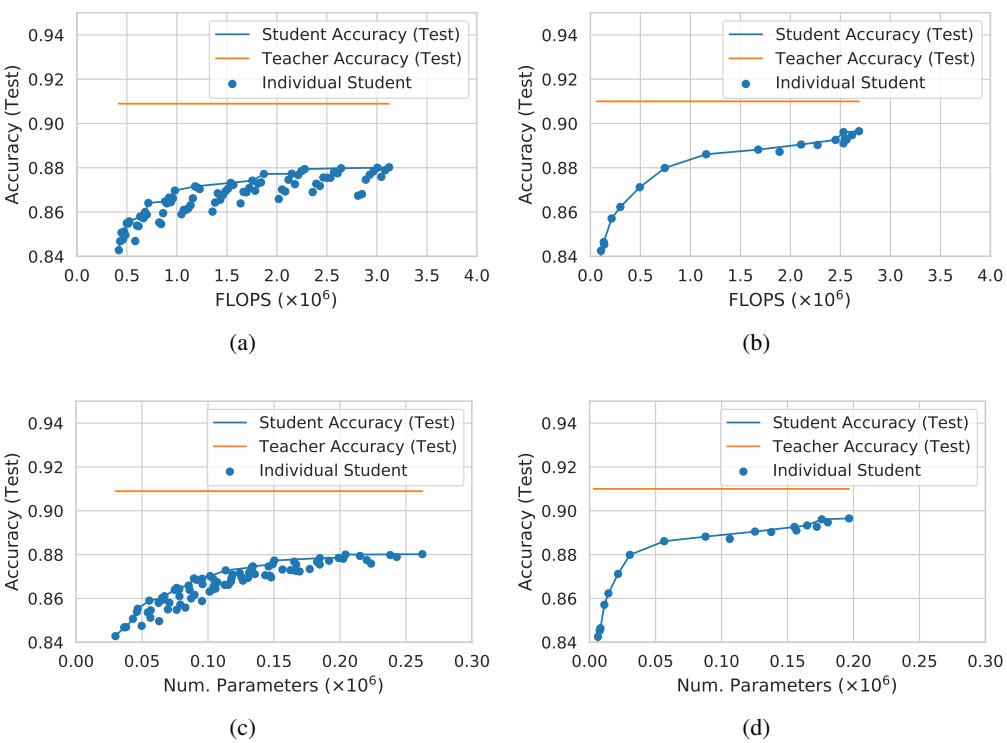

Figure 5: Accuracy-Storage-Computation tradeoff while using CNNs on MNIST with masking rate 29%. (a) Test accuracy using posterior predictive distribution vs FLOPS found using exhaustive search. (b) Test accuracy using posterior predictive distribution vs FLOPS found using group $\ell_1/\ell_2$ with pruning. (c) Test accuracy using posterior predictive distribution vs storage found using exhaustive search. (d) Test accuracy using posterior predictive distribution vs storage found using group $\ell_1/\ell_2$ with pruning. The optimal student model for this configuration is obtained with group $\ell_1/\ell_2$ pruning. It has approximately $6.6\times$ the number of parameters and $6.4\times$ the FLOPS of the base student model.

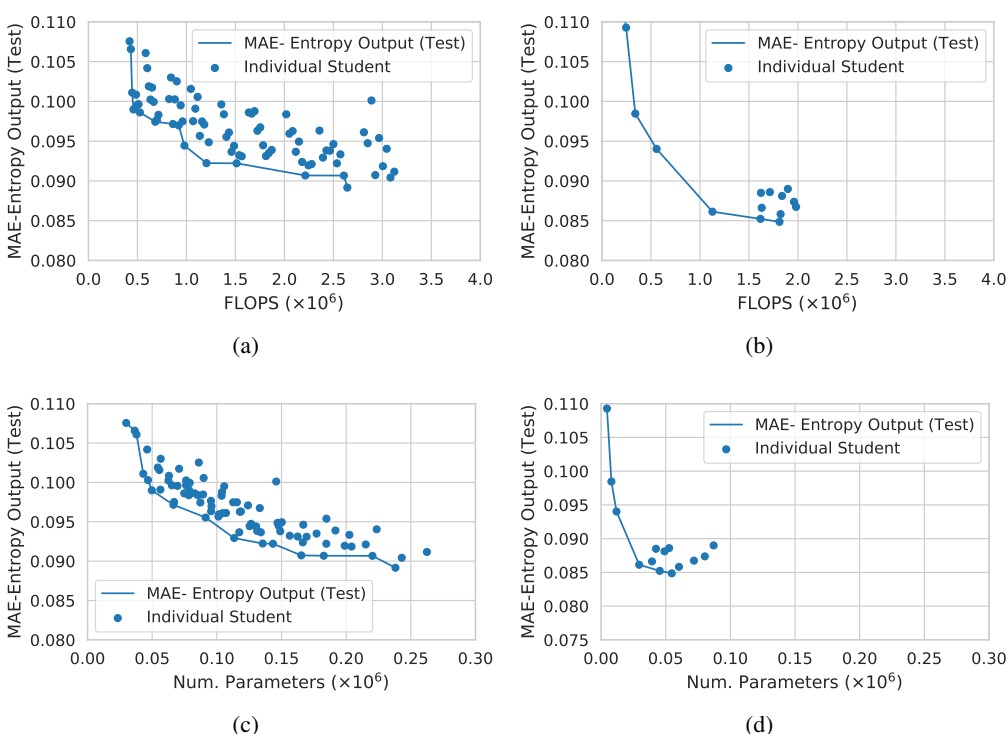

Figure 6: Entropy Error-Storage-Computation tradeoff while using CNNs on MNIST with masking rate $29\%$. (a) Test mean absolute error for posterior entropy vs FLOPS found using exhaustive search. (b) Test mean absolute error for posterior entropy vs FLOPS found using group $\ell_1/\ell_2$ with pruning. (c) Test mean absolute error for posterior entropy vs storage found using exhaustive search. (d) Test mean absolute error for posterior entropy vs storage found using group $\ell_1/\ell_2$ with pruning. The optimal student model for this configuration is obtained with group $\ell_1/\ell_2$ pruning. It has approximately $1.8\times$ the number of parameters and $4.3\times$ the FLOPS of the base student model.

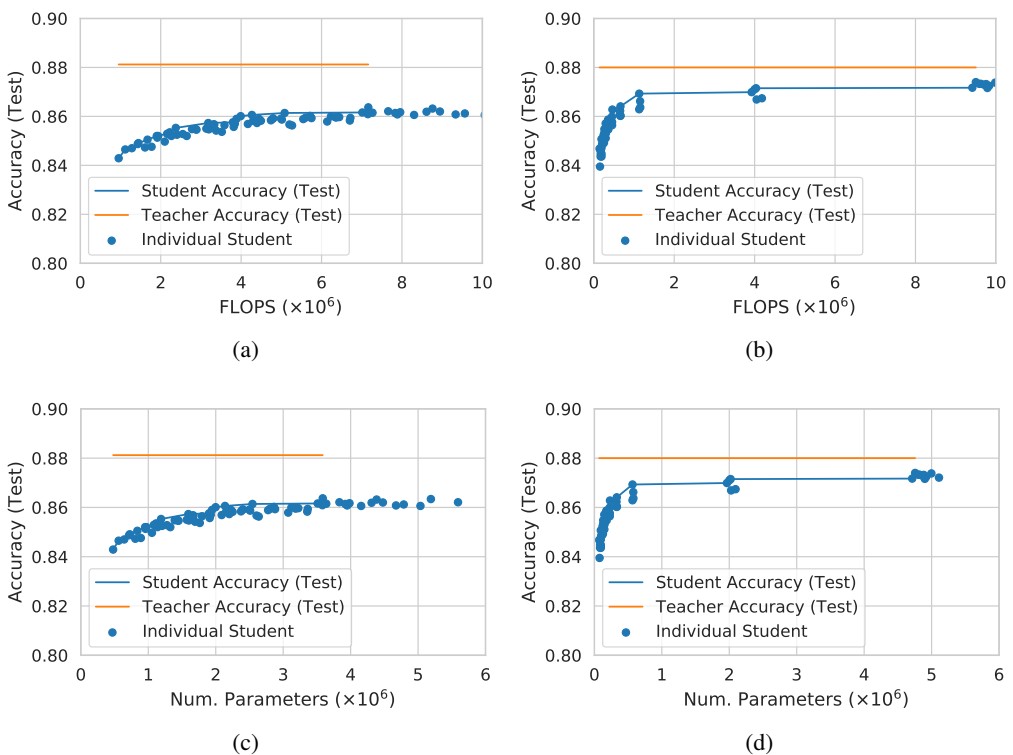

Figure 7: Accuracy-Storage-Computation tradeoff while using Fully-connected networks on MNIST with masking rate 29%. (a) Test accuracy using posterior predictive distribution vs FLOPS found using exhaustive search. (b) Test accuracy using posterior predictive distribution vs FLOPS found using group $\ell_1/\ell_2$ with pruning. (c) Test accuracy using posterior predictive distribution vs storage found using exhaustive search. (d) Test accuracy using posterior predictive distribution vs storage found using group $\ell_1/\ell_2$ with pruning. The optimal student model for this configuration is obtained with group $\ell_1/\ell_2$ pruning. It has approximately $9.9\times$ the number of parameters and $10\times$ the FLOPS of the base student model.

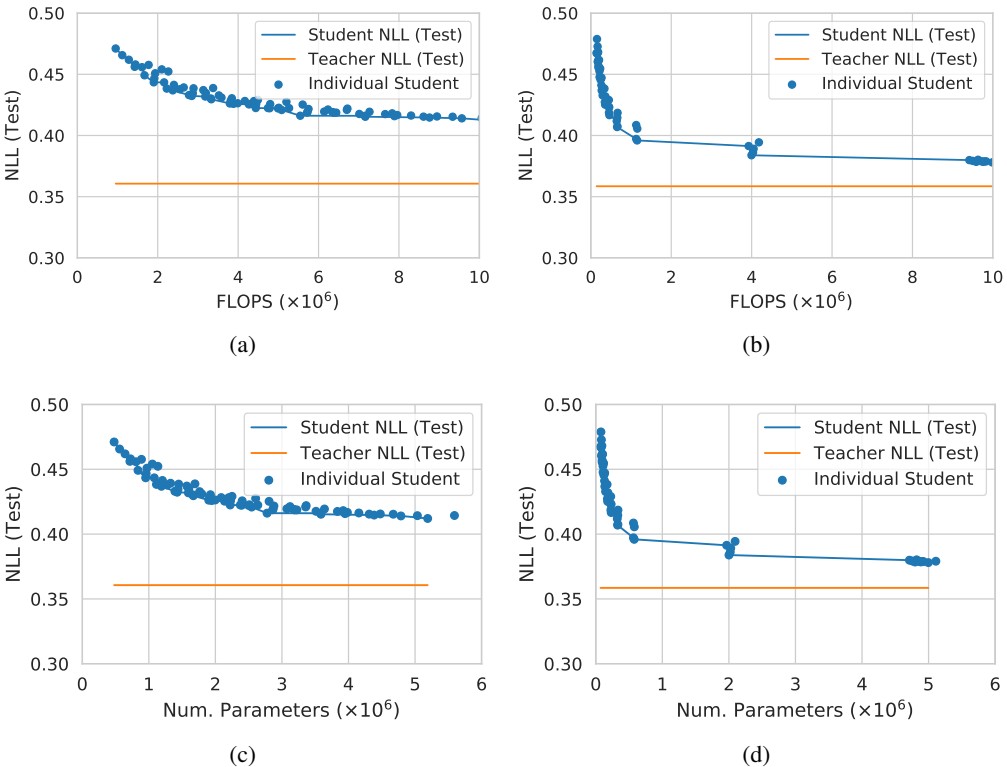

Figure 8: NLL-Storage-Computation tradeoff while using Fully-connected networks on MNIST with masking rate 29%. (a) Test negative log likelihood of posterior predictive distribution vs FLOPS found using exhaustive search. (b) Test negative log likelihood of posterior predictive distribution vs FLOPS found using group $\ell_1/\ell_2$ with pruning. (c) Test negative log likelihood of posterior predictive distribution vs storage found using exhaustive search. (d) Test negative log likelihood of posterior predictive distribution vs storage found using group $\ell_1/\ell_2$ with pruning. The optimal student model for this configuration is obtained with group $\ell_1/\ell_2$ pruning. It has approximately $9.9\times$ the number of parameters and $10\times$ the FLOPS of the base student model.

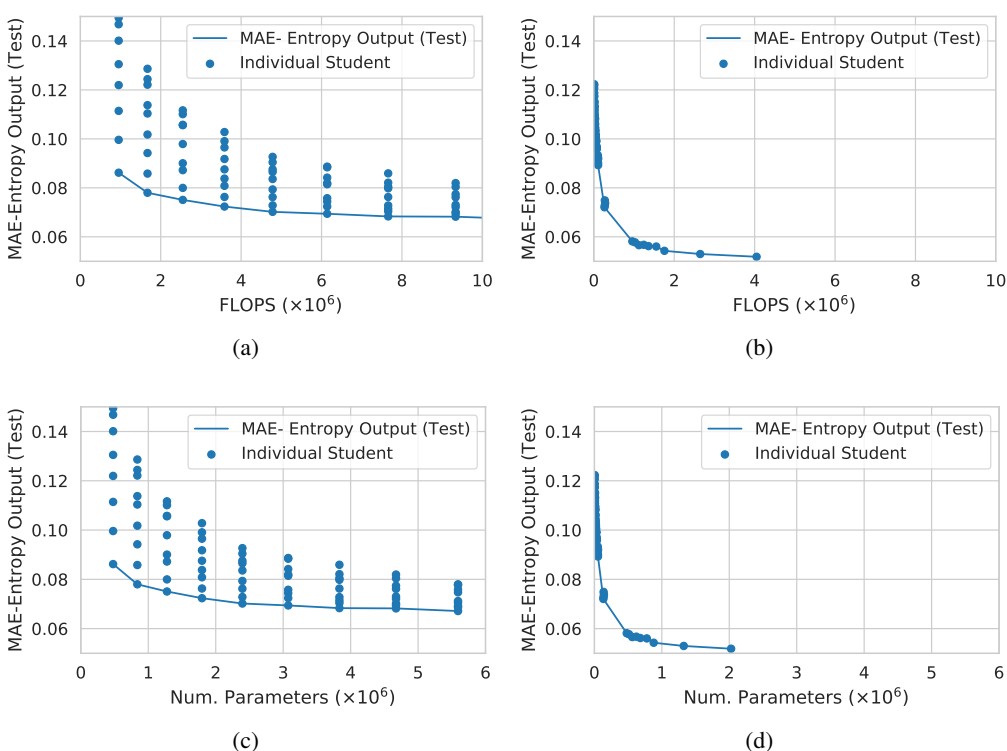

Figure 9: Entropy Error-Storage-Computation tradeoff while using Fully-connected networks on MNIST with masking rate 29%. (a) Test mean absolute error for posterior entropy vs FLOPS found using exhaustive search. (b) Test mean absolute error for posterior entropy vs FLOPS found using group $\ell_1/\ell_2$ with pruning. (c) Test mean absolute error for posterior entropy vs storage found using exhaustive search. (d) Test mean absolute error for posterior entropy vs storage found using group $\ell_1/\ell_2$ with pruning. The optimal student model for this configuration is obtained with group $\ell_1/\ell_2$ pruning. It has approximately $4.2\times$ the number of parameters and $4.2\times$ the FLOPS of the base student model.

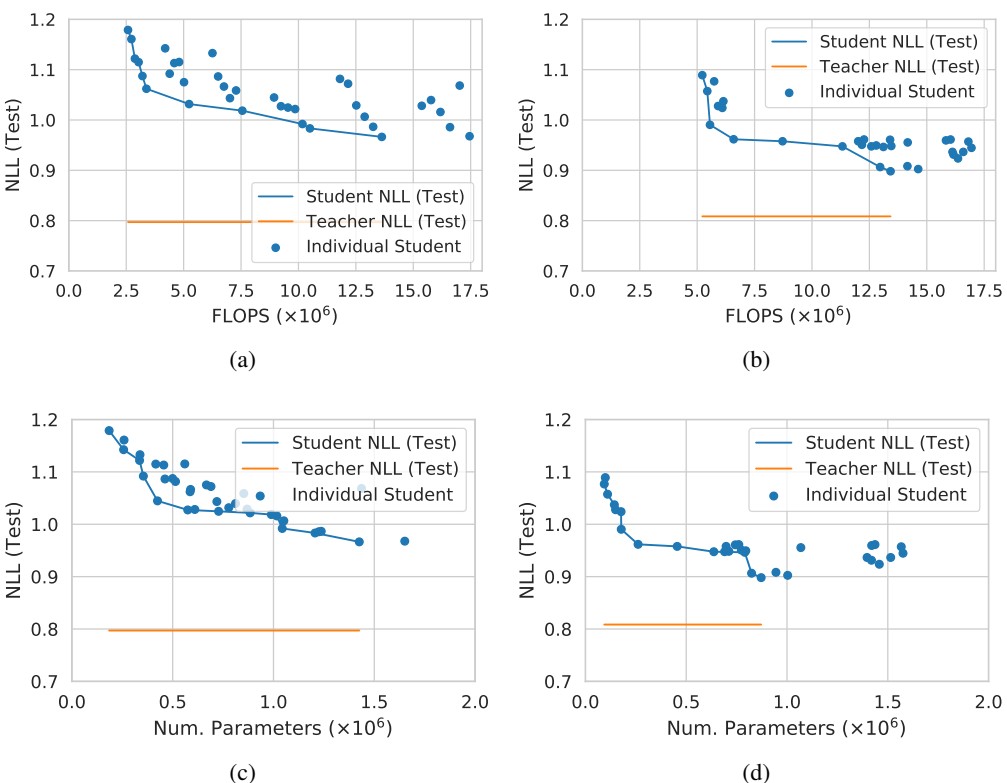

Figure 10: NLL-Storage-Computation tradeoff while using CNNs on CIFAR10 with training set size of 20,000 samples. (a) Test negative log likelihood of posterior predictive distribution vs FLOPS found using exhaustive search. (b) Test negative log likelihood of posterior predictive distribution vs FLOPS found using group $\ell_1/\ell_2$ with pruning. (c) Test negative log likelihood of posterior predictive distribution vs storage found using exhaustive search. (d) Test negative log likelihood of posterior predictive distribution vs storage found using group $\ell_1/\ell_2$ with pruning. The optimal student model for this configuration is obtained with group $\ell_1/\ell_2$ pruning. It has approximately $4.7\times$ the number of parameters and $5.2\times$ the FLOPS of the base student model.

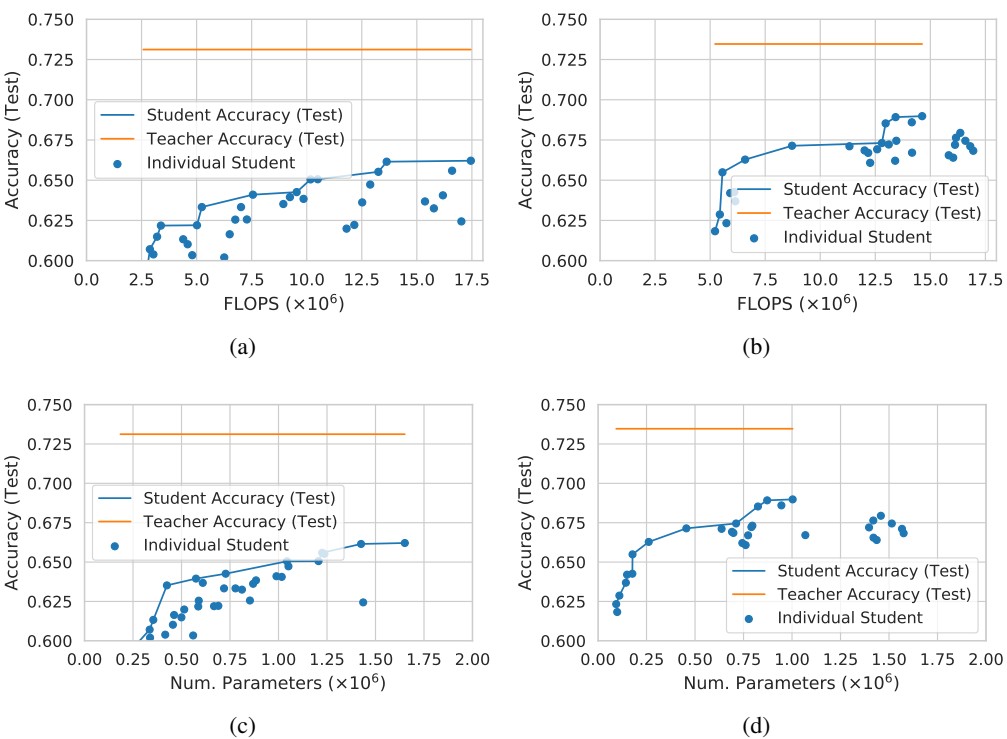

Figure 11: Accuracy-Storage-Computation tradeoff while using CNNs on CIFAR10 with sub-sampling training data to 20,000 samples. (a) Test accuracy using posterior predictive distribution vs FLOPS found using exhaustive search. (b) Test accuracy using posterior predictive distribution vs FLOPS found using group $\ell_1/\ell_2$ with pruning. (c) Test accuracy using posterior predictive distribution vs storage found using exhaustive search. (d) Test accuracy using posterior predictive distribution vs storage found using group $\ell_1/\ell_2$ with pruning. The optimal student model for this configuration is obtained with group $\ell_1/\ell_2$ pruning. It has approximately $5.4\times$ the number of parameters and $5.6\times$ the FLOPS of the base student model.

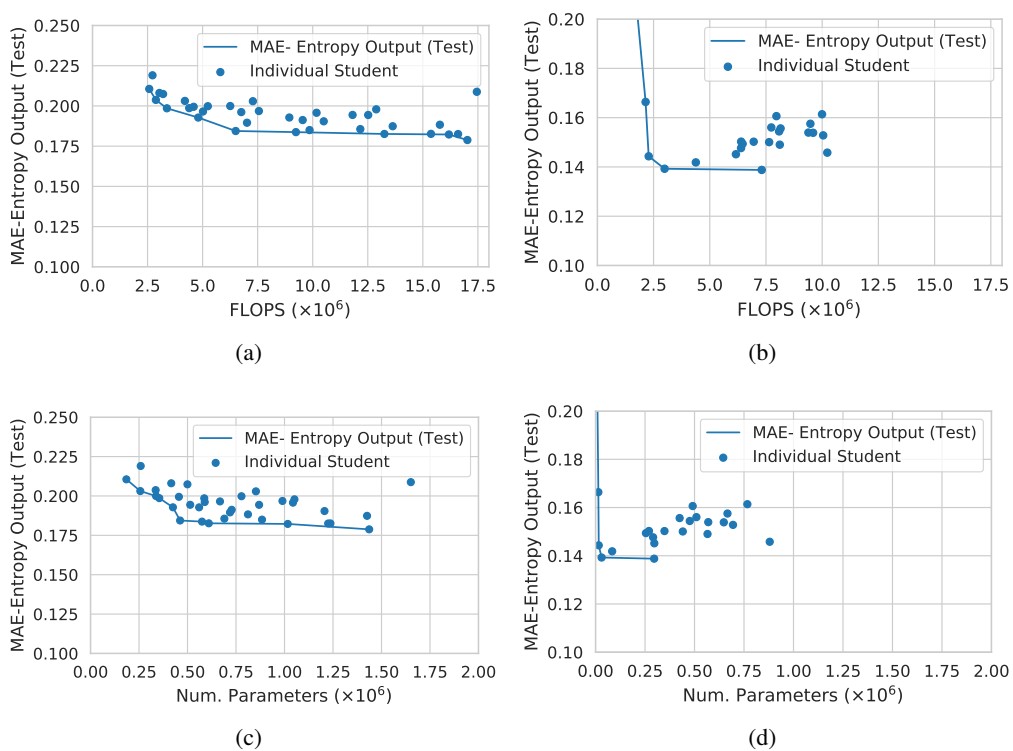

Figure 12: Entropy Error-Storage-Computation tradeoff while using CNNs on CIFAR10 with subsampling training data to 20,000 samples. (a) Test mean absolute error for posterior entropy vs FLOPS found using exhaustive search. (b) Test mean absolute error for posterior entropy vs FLOPS found using group $\ell_1/\ell_2$ with pruning. (c) Test mean absolute error for posterior entropy vs storage found using exhaustive search. (d) Test mean absolute error for posterior entropy vs storage found using group $\ell_1/\ell_2$ with pruning. The optimal student model for this configuration is obtained with group $\ell_1/\ell_2$ pruning. It has approximately $1.6\times$ the number of parameters and $2.8\times$ the FLOPS of the base student model.

