# OpenReview forum: "Generalized Bayesian Posterior Expectation Distillation for Deep Neural Networks"
_ICLR.cc/2020/Conference — Reject_

### Official Review · AnonReviewer3 · 2019-10-17
**Official Blind Review #3**

**Rating:** 3

**Review:**

Contributions:

The paper considers the distillation of a Bayesian neural network as presented in [Balan et al. 2015]

The main contribution of the paper is the extension of [Balan et al. 2015] to apply to general posterior expectations instead of being restricted to predictions.

A second contribution of the paper is the finding that restricting the architecture of the student network to coincide with the teacher can lead to suboptimal performance and this can be mitigated by expanding the student's architecture using architecture search.

Originality/Significance:

I want to discuss the result regarding the generalization of the posterior expectation (section 3.1). To my knowledge this is novel, however, I am failing to see the importance of this result. The paper mentions two cases as motivations: to calculate the entropy and the variance of the marginal. The  problem with these two examples is that it is unclear why these are important and they are not used in the experiments anywhere. Their use should be motivated and the performance of the distillation should be properly evaluated in the experiments.

The result regarding architecture search is interesting, but it should be expanded and explained more to be a main contribution.

Clarity:

The paper generally understandable, and well written, but it could be better organized. It should expand on the motivation and the architecture search, since these are key components of the paper. The figures are not legible. Even fully zoomed in, they are difficult to read.

Overall assessment:

The paper has some interesting ideas, but it lacks motivation and significant results.

_______________________________________________________________________

Response to the rebuttal:

Thank you for the detailed reply.

> A primary motivation for generalising posterior expectations is to help quantify model uncertainty. Indeed, the expectation of the predictive entropy is an important quantity that is distinct from the entropy of the posterior predictive distribution. For instance, the difference between these two quantities is exactly the BALD score used in active learning [1]. ...

BALD would be problematic to use with this framework due to computational costs. The main benefit of distillation is the reduced computational cost at inference time. But training itself is still expensive. In BALD, the bulk of the computational cost is fitting the model after each new observation. Distillation does not provide a speedup here.

If the method indeed works well in an active learning setting, it would be interesting to see experiments showcasing this result.

> - The result regarding architecture search is interesting, but it should be expanded and explained more to be a main contribution.

I was hoping for more discussion/guidance on finding the right architecture or perhaps an algorithm that efficiently optimises the architecture. But I understand that this is more of a future work so I am not holding this against the paper.

I realise that my initial assessment was rather short so I decided to increase my rating and lower my confidence score. I think the paper is borderline, but I am slightly leaning towards rejection due to the insufficient motivation.

**Experience Assessment:**

I have published one or two papers in this area.

**Review Assessment: Checking Correctness Of Derivations And Theory:**

I did not assess the derivations or theory.

**Review Assessment: Checking Correctness Of Experiments:**

I did not assess the experiments.

**Review Assessment: Thoroughness In Paper Reading:**

I read the paper at least twice and used my best judgement in assessing the paper.

---

> ### Author Response · Authors · 2019-11-08
> **Response to Review#3**
>
> Thank you for taking the time out to provide feedback on our work. Below we address the concerns raised in the review:
>
>
> - I want to discuss the result regarding the generalization of the posterior expectation (section 3.1). To my knowledge this is novel, however, I am failing to see the importance of this result. The paper mentions two cases as motivations: to calculate the entropy and the variance of the marginal. The problem with these two examples is that it is unclear why these are important and they are not used in the experiments anywhere. Their use should be motivated and the performance of the distillation should be properly evaluated in the experiments.
>
> > First, we note that we have indeed used the expectation of the predictive entropy under the posterior as a distillation target in our experiment. These results are in Section 4. Table 1 (last column). Further,  Figure 1(c, f) and Figure 4(c) show the performance of the student model on distilling entropy as we apply our data augmentation techniques to vary posterior uncertainty. Finally, in the appendix, Figures 6, 9, 12 show the error-storage-computation tradeoff for entropy for the student model whose architecture is determined by either exhaustive search and group l1/l2 pruning method for different model-dataset combinations.
>
> > A primary motivation for generalizing posterior expectations is to help quantify model uncertainty. Indeed, the expectation of the predictive entropy is an important quantity that is distinct from the entropy of the posterior predictive distribution. For instance, the difference between these two quantities is exactly the BALD score used in active learning [1]. Note that the entropy of the posterior predictive distribution can be high while the expectation of the predictive entropy can be low, indicating that there are multiple competing hypotheses that each assign high confidence to an instance. This structure of the posterior can not be revealed without the expectation of the predictive entropy. Our proposed distillation approach then yields a computationally efficient method to compute both of these quantities.
>
> - The result regarding architecture search is interesting, but it should be expanded and explained more to be a main contribution.
>
> > We are a little unclear as to what the concern is in this comment. It’ll help us address this concern better if you could elaborate a bit more on this.
>
> - The paper generally understandable, and well written, but it could be better organized. It should expand on the motivation and the architecture search, since these are key components of the paper. The figures are not legible. Even fully zoomed in, they are difficult to read.
>
> > As mentioned in our previous response above, it’ll be very helpful to us if you could elaborate on the feedback regarding the explanation around architecture search. Lastly, we have updated the paper to ensure that all figures are legible at print resolution.
>
> References:
> [1] Neil Houlsby, Ferenc Huszár, Zoubin Ghahramani, and Máté Lengyel. Bayesian active learning for classification and preference learning. arXiv preprint arXiv:1112.5745, 2011.

---

> > ### Author Response · Authors · 2019-11-15
> > **Requesting clarification on the architecture search comment**
> >
> > As the rebuttal period is ending, we would like to note that we would be happy to address your concerns about the architecture search section of the paper if you can provide more detail in your final review.

---

### Official Review · AnonReviewer1 · 2019-10-23
**Official Blind Review #1**

**Rating:** 6

**Review:**

The authors consider the problem of distilling expectations with respect to Bayesian neural network (BNN) posteriors. These expectations rely on Monte Carlo integration and owing to the large number of BNN parameters can be computationally expensive and memory intensive to compute, motivating the need for distillation.

I recommend a weak accept for the paper. The authors generalize previous work on distilling posterior predictives by allowing for the computation of posterior expectations beyond posterior predictive distributions, proposing alternate low variance MC estimators, and using an amortization network whose architecture need not be identical to the original BNNs architecture.

While the extensions individually are incremental and not particularly exciting, taken together, I believe, they do address a gap in the existing literature. The experiments successfully demonstrate a) when naive distillation fails and b) the proposed extensions help alleviate some of the observed issues. The paper would likely be an useful resource for practitioners in the area.

Minor:
+ Us vs Uo estimators: It would be interesting to more clearly see what the additional storage (and computation) of Uo is buying us. How much worse are the posterior predictive entropies if Uo is switched with Us? And do the posterior predictive estimates improve if Uo is used inlace of Us?

+ In the paragraph following equation 4, the posterior marginal variance expression implicitly assumes that p(y|x, \theta) is a Categorical distribution. This should be clarified. The expression doesn’t generally hold, for example if p(y | x, \theta) is a Gaussian.

+ Figures 1 and 2 are too small and difficult to parse. I would recommend moving some of these to the supplement.

+ It would be good to explicitly point out how much larger is the best (one with the smallest teacher student gap) l1/l2 regularized model compared to the base student model. I realize this is hiding in Figure 2 somewhere, but is not obvious.

**Experience Assessment:**

I have published in this field for several years.

**Review Assessment: Checking Correctness Of Derivations And Theory:**

N/A

**Review Assessment: Checking Correctness Of Experiments:**

I assessed the sensibility of the experiments.

**Review Assessment: Thoroughness In Paper Reading:**

I read the paper at least twice and used my best judgement in assessing the paper.

---

> ### Author Response · Authors · 2019-11-07
> **Response to Review#1**
>
> Thank you for spending time on our paper and for giving your review. Below, we address some of the concerns raised in the review:
>
> + In the paragraph following equation 4, the posterior marginal variance expression implicitly assumes that p(y|x, \theta) is a Categorical distribution. This should be clarified. The expression doesn’t generally hold, for example if p(y | x, \theta) is a Gaussian.
>
> > In this work, we focus on the classification setting explicitly (see, for example, the fourth line of the abstract that establishes this framing as well as the  start of Section 3). As such, we assume throughout that p(y|x, \theta) is categorical.
>
> + Figures 1 and 2 are too small and difficult to parse. I would recommend moving some of these to the supplement.
>
> > Thank you for noting this issue. We have updated the paper to ensure that all figures are legible at print resolution.
>
> + It would be good to explicitly point out how much larger is the best (one with the smallest teacher student gap) l1/l2 regularized model compared to the base student model. I realize this is hiding in Figure 2 somewhere, but is not obvious.
>
> > Interpreting the size of a model in terms of the number of parameters, the CNN case for MNIST posterior predictive distribution distillation results in the best model under group l1/l2 pruning that has roughly 6.6 times the number of parameters when compared to the base student model. We have updated the figure captions to include this information.
>
> We are also thankful for the additional comments in your review. We will respond to them subsequently.

---

> > ### Author Response · Authors · 2019-11-12
> > **[Contd.] Response to Review#1**
> >
> > Thank you for your patience. Our response to your additional comment is given below.
> >
> > + Us vs Uo estimators: It would be interesting to more clearly see what the additional storage (and computation) of Uo is buying us. How much worse are the posterior predictive entropies if Uo is switched with Us? And do the posterior predictive estimates improve if Uo is used in place of Us?
> >
> > > The additional storage and computation required by the Uo estimator will be O(N), where N is the total number of cases in our training set used for training the student. To illustrate the effect of using Uo vs Us, we ran additional experiments on CIFAR10 with the same experimental configurations used in Figure 1(f). The results are reported below:
> >
> > Number of training samples for teacher | MAE (Entropy) - Uo | MAE (Entropy) - Us
> >                    10000                                           |             0.144              |             0.192
> >                    20000                                           |             0.210              |             0.231
> >                    50000                                           |             0.245               |             0.290
> >
> > Note that the results for Uo in the table above are already present in Figure 1(f). As we can see, Uo achieves a lower mean absolute error (about 15% - 25% lower) while distilling expectation of entropy. We will also run the same set of experiments for both the models on MNIST and include them in the supplemental section of the paper.

---

### Official Review · AnonReviewer2 · 2019-10-26
**Official Blind Review #2**

**Rating:** 6

**Review:**

Summary:
The paper introduces a general framework for distilling expectations of the Bayesian posterior distribution of a deep neural network, aiming to extend the original Bayesian Dark Knowledge approach [1]. More concretely, the generalized framework takes as input a teacher network, a general posterior expectation of interest, a student network, and thus performs an online compression of the selected posterior expectation using iteratively generated Monte Carlo samples from the parameter posterior of the teacher model. The proposed framework is applied to the case of classification models and empirical results demonstrate that distilling into a student model with an architecture that matches the teacher, as is done in Bayesian Dark Knowledge, can lead to sub-optimal performance. It is also shown that student architecture search methods can identify student models with significantly improved speed-storage-accuracy trade-offs.

Strengths:
Overall, the paper is well written and the relationship to previous works is well described. I personally like the Bayesian Dark Knowledge approach, which combines SGLD and knowledge distillation or dark knowledge, and very happy to see its generalization. Unlike the previous work, it is clearly shown that restricting the student architecture to match the teacher can sometimes lead to a significant performance drop, which provides a basis for guiding future developments.

Weaknesses:
- I think it is a valuable contribution, but my major concern is that the authors only conduct experiments for the classification task, whereas the original Bayesian Dark Knowledge approach also deals with the regression task and shows some interesting results (see Sect. 3.2 and 3.3 in Ref. [1]). I would recommend the authors to extend the experimental evaluation and provide some insight on how to extend the proposed framework to cover the regression task.
- On page 5, the choice of loss function does not seem to be discussed. I would like the authors to clarify why cross entropy loss is replaced with l(h, h’)=|h-h’| in the classification case.
- The size of some figures appears too small, for example Fig. 1 and Fig. 2, which may hinder readability.

At the moment, I recommend a weak reject as the main weakness is the experimental evaluation, but I could be open to increasing my score if my concerns are addressed.

References:
[1] Anoop Korattikara Balan, Vivek Rathod, Kevin P Murphy, and Max Welling. Bayesian dark knowledge. In Advances in Neural Information Processing Systems, pp. 3438–3446, 2015.

**Experience Assessment:**

I have read many papers in this area.

**Review Assessment: Checking Correctness Of Derivations And Theory:**

I assessed the sensibility of the derivations and theory.

**Review Assessment: Checking Correctness Of Experiments:**

I assessed the sensibility of the experiments.

**Review Assessment: Thoroughness In Paper Reading:**

I read the paper at least twice and used my best judgement in assessing the paper.

---

> ### Author Response · Authors · 2019-11-07
> **Response to Review#2**
>
> Thank you for taking the time out to provide feedback on our work. Below, we address some of the concerns raised in your review:
>
> - On page 5, the choice of loss function does not seem to be discussed. I would like the authors to clarify why cross-entropy loss is replaced with l(h, h’)=|h-h’| in the classification case.
>
> > The cross-entropy loss is used when we are distilling the expectation of the predictive distribution E[p(y|x, \theta)] under the parameter posterior p(\theta|D). We switch to l(h, h’)=|h-h’| when we are distilling the expectation of the predictive entropy E[H[y|x,D, \theta]] under the parameter posterior. Note that the distillation problem for posterior entropy is actually a regression problem and the use of the absolute loss to assess performance on this task is intuitive as it measures the absolute difference in entropy in units of nats.
>
> - The size of some figures appears too small, for example, Fig. 1 and Fig. 2, which may hinder readability.
>
> > Thank you for noting this issue. We have updated the paper to ensure that all figures are legible at print resolution.
>
> We are also very grateful for the additional comments in the review. We will be responding to them subsequently.

---

> > ### Author Response · Authors · 2019-11-12
> > **[Contd.] Response to Review#2**
> >
> > Thank you for your patience. Our response to your additional comment is given below.
> >
> > - I think it is a valuable contribution, but my major concern is that the authors only conduct experiments for the classification task, whereas the original Bayesian Dark Knowledge approach also deals with the regression task and shows some interesting results (see Sect. 3.2 and 3.3 in Ref. [1]). I would recommend the authors to extend the experimental evaluation and provide some insight on how to extend the proposed framework to cover the regression task.
> >
> > > We agree with the reviewer that the regression results in the original Bayesian Dark Knowledge paper [1] are interesting. However, the current paper is explicitly framed to focus only on the classification task to enable a much deeper exploration of the method than was performed in the original paper. Indeed, this exploration has revealed fundamental issues with the assumptions of the original method (e.g., that it is possible to compress the posterior for a network back into a student network of the same size), and we have provided and evaluated solutions to this problem via methods to configure the student architecture. We have also extended the original work in a different direction by studying the distillation of generalized posterior expectations specifically in the classification setting. Together, we feel that our existing results as presented in the current paper will provide significant value to the community.
> >
> > References:
> > [1] Anoop Korattikara Balan, Vivek Rathod, Kevin P Murphy, and Max Welling. Bayesian dark knowledge. In Advances in Neural Information Processing Systems, pp. 3438–3446, 2015.

---

### Author Response · Authors · 2019-11-15
**Revision Summary**

Based on the useful feedback given by the reviewers, we have made multiple revisions to the paper. Below we summarize the changes between the version that was reviewed and the final version that we have now.

1) There were concerns about the readability of the figures. We have updated the paper to ensure that all figures are legible at print resolution.

2) Reviewer #1 wanted us to report the performance comparison between Uo and Us for different targets and also highlight the additional storage and computation cost required for Uo. We ran additional experiments to compare Uo and Us for both the distillation targets on all model-data set combinations. The results from these experiments are present in the Appendix (Tables 2-4). We have also mentioned explicitly in our paper that the additional storage and computation requirement for Uo grows linearly in the number of training set instances available for the student model.

3) Reviewer #3 felt that the motivation behind choosing our distillation targets was not explained clearly. We have responded to reviewer #3 and also updated our introduction section to expand on the motivation.

4) Reviewer #1 also suggested mentioning explicitly how large the optimal student model was after architecture search w.r.t the base student model. We have updated all the figures which had the performance-computation-storage tradeoff plots to include this information.

In addition to these revisions, we have also addressed the concerns of each of the reviewers in the individual responses posted.

---

### Decision · Program_Chairs · 2019-12-19

**Decision:**

Reject

**Comment:**

The authors consider distilling posterior expectations for Bayesian neural networks. While reviewers found the material interesting, and the responses thoughtful, there were questions about the practical utility of the work. Evaluations of classification favour NLL (and typically do not show accuracy), and regression (which was considered in the original Bayesian Dark Knowledge paper) is not considered. In general, it is difficult to assess and interpret how the approach is working, and in what application regime it would be a gold standard, e.g., with respect to downstream tasks. The authors are encouraged to continue with this work, taking reviewer comments into account in a final version.